# OSS-BENCH: EVALUATING LLMS VIA THE REALISTIC OPEN-SOURCE SOFTWARE DEVELOPMENT PIPELINE

## ABSTRACT

In light of the rapid adoption of AI coding assistants, LLM-assisted development has become increasingly prevalent, creating an urgent need for robust evaluation of generated code quality. Existing benchmarks often require extensive manual effort to create static datasets, rely on indirect or insufficiently challenging tasks, depend on non-scalable ground truth, or neglect critical low-level security evaluations, particularly memory-safety issues. In this work, we introduce OSS-BENCH, a benchmark generator that automatically constructs large-scale, live evaluation tasks from real-world Open-Source Software (OSS). OSS-BENCH replaces functions with LLM-generated code and evaluates them using three novel metrics: compilability, functional correctness, and memory safety, leveraging robust signals like compilation failures, test-suite violations, and sanitizer alerts as ground truth. In our evaluation, the benchmark, instantiated as OSS-BENCH$_{php}$ and OSS-BENCH$_{sql}$, profiles 17 diverse LLMs at million-scale tasks, revealing insights such as intra-family behavioral patterns and inconsistencies between model size and performance. Our results demonstrate that OSS-BENCH mitigates overfitting by leveraging the evolving complexity of OSS and highlight LLMs' limited understanding of low-level code security via extended fuzzing experiments. Overall, OSS-BENCH offers a practical and scalable framework for benchmarking the real-world coding capabilities of LLMs.

## 1 INTRODUCTION

Coding with Large Language Model (LLM) assistants such as Copilot (GitHub, Inc., 2021) and Cursor (Cursor Team, 2023) is revolutionizing software development by enabling users to refine code snippets through natural language. Numerous coding LLMs have emerged in recent years, each claiming strong capabilities across various programming tasks and supporting these claims with competitive or state-of-the-art benchmark results. This rapid growth has created an urgent need for rigorous and robust evaluation of the quality of LLM-generated code.

Numerous benchmarks (Liu et al., 2023; Hajipour et al., 2024; Hu et al., 2025; Zheng et al., 2025; Wang et al., 2025; Tang et al., 2023; Jimenez et al., 2023a; Jain et al., 2024; Zhuo et al., 2024; Zan et al., 2025; Dilgren et al., 2025; Peng et al., 2025; Zheng et al., 2024b; Baars & Meester, 2019; Quan et al., 2025; Feng et al., 2024; Yadav & Singh, 2024; Zheng et al., 2024a; Jimenez et al., 2023b; Du et al., 2023; Peng et al., 2024; Zheng et al., 2023) have been proposed to assess and quantify LLM coding capabilities, typically focusing on three key aspects: correctness, efficiency, and security. While these benchmarks provide valuable insights into the coding abilities of various LLMs, they commonly suffer from at least one of the following limitations:

- *Heavy human effort and static tasks.* Many benchmarks rely on manually crafted test cases, requiring substantial human expertise and effort. These datasets are typically static and lack extensibility, potentially allowing models to overfit by memorizing or hard-coding solutions.

- *Indirect or less challenging tasks.* Some benchmarks use indirect prompting (*e.g.,* needle-searching tasks) or simple coding scenarios, such as basic algorithm implementations, which do not sufficiently challenge rapidly evolving LLMs.

- *Limited or non-scalable ground truth.* Manually constructed benchmarks are costly, subjective, error-prone, and fail to scale. Automated alternatives, such as using other LLMs to generate ground truth, introduce additional errors and uncertainties into the evaluation.
- *Limited focus on code security.* Most existing benchmarks prioritize correctness/efficiency, but not much attention has been paid to security. In particular, low-level code vulnerabilities (*e.g.,* memory-safety issues) remain largely unexplored in current LLM evaluation frameworks.

We advocate for a fully automated approach to evaluating the coding capabilities of LLMs, without human or LLM involvement in test case generation or ground truth annotation, while preserving practically relevant, sufficiently challenging programming tasks that span a diverse range of real-world code. In this work, we introduce **OSS-BENCH**, an automatic benchmark approach designed to address the challenges outlined above without manual intervention or LLM-generated oracles. To reflect real-world development practices, OSS-BENCH prompts LLMs to perform function-level code edits, a common task for developers.[1] To enhance task complexity, it draws from the rich and mature ecosystem of large-scale OSS projects, encompassing sophisticated components such as garbage collectors, compilers, and memory allocators. Inspired by the practical software pipeline that compiles sources, runs tests, and checks safety, OSS-BENCH derives robust ground truth from three practical, verifiable metrics as follows:

- **Compilability.** This measures whether a modified function compiles successfully; failures signal syntax errors (*e.g.,* invalid keywords) or semantic errors (*e.g.,* references to non-existent variables).
- **Functional Test.** Software testing serves as a proxy for code correctness. Well-maintained OSS projects typically include comprehensive test suites (such as unit tests, integration tests, end-to-end tests and regression tests) to validate the functionality of code changes.
- **Memory Safety.** Memory safety is evaluated through sanitizer tools (*e.g.,* ASan (Serebryany et al., 2012)) that detect memory errors such as buffer overflows and use-after-free that can be attacked or exploited.

We instantiate OSS-BENCH to generate two real-world benchmarks, OSS-BENCH$_{php}$ and OSS-BENCH$_{sql}$, built on the PHP (The PHP Group, 2025) interpreter and the SQLite3 (Hipp, 2025) engine. Both benchmarks prompt LLMs to **improve** individual functions and then measure the **degradation** (*i.e.,* new compilation failures, test regressions, or sanitizer alerts in the original code) after replacing the original function with the LLM version. OSS-BENCH delivers (1) continuously updatable tasks with no manual curation, (2) complex, realistic challenges grounded in natural OSS signals without relying on other LLMs, and (3) measurements of low-level memory safety.

We assess OSS-BENCH 's effectiveness and fairness on both OSS-BENCH$_{php}$ and OSS-BENCH$_{sql}$ using 17 widely used LLMs. Our results show that OSS-BENCH not only profiles compilation, testing, and memory-safety performance accurately, but also uncovers insightful observations, such as inconsistencies between model size and performance, and how increased memorization can lead to counterproductive outputs. Furthermore, we demonstrate that OSS-BENCH 's live, continually updated design mitigates overfitting to fixed datasets and scales naturally to large, evolving codebases.

## 2 APPROACH

**Benchmark Workflow.** Figure 1 illustrates OSS-BENCH 's four-step workflow: (I) select an OSS project, extract function-level snippets, and prompt LLMs for optimized versions; (II) evaluate the compilability of each LLM-generated function; (III) sample the successfully compiled functions to form testing datasets and run the OSS test suite; and (IV) collect compilation and test logs and analyse sanitizer alerts for memory-safety assessment.

Steps (II), (III), and (IV) correspond to three core metrics for evaluating LLM-generated code quality. A model receives a higher compilation score if more LLM modifications compile successfully; a higher test score if the average test pass rate across sampled iterations is higher; and a higher memory safety score if it triggers fewer sanitizer alerts. In principle, users can apply this workflow to various open-source projects to create diverse benchmarks in different areas.

---

[1]The task and prompt engineering are flexible. We highlight the contributions of introducing the realistic development pipeline with three novel and robust metrics.

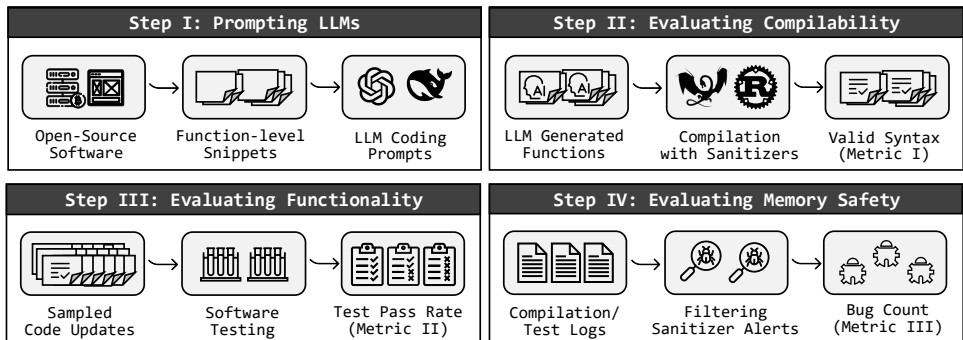

Figure 1: Benchmark Workflow in OSS-BENCH

**Open-source Software Criteria.** Decades-old, actively maintained open-source software provides an ideal foundation for live benchmarks. Consequently, to select projects that offer substantial code complexity, extensive test coverage, and stable release practices, we apply the following criteria:

- *Large codebase.* A large codebase increases the inherent complexity of evaluation tasks on realistic development scenarios. Although the LLM edits a small code snippet, that snippet must integrate seamlessly into the broader, complex codebase.
- *Comprehensive test suite.* The selected OSS must include an actively maintained, well-designed test suite with broad coverage (*e.g.,* 60% code coverage, the acceptable coverage considered by Google (Google, 2020)). Ideally, the suite combines unit tests and end-to-end tests to validate both individual functions and their interactions across the codebase.
- *Compiled languages.* Compiled programming languages (*e.g.,* C, C++, Rust) that require a build step are inherently more complex than scripting languages (*e.g.,* Python, JavaScript). Evaluating on compiled languages (1) increases coding task difficulty and (2) provides natural, robust ground truth via compilability and sanitizer checks.

Given these criteria, two major OSS categories commonly qualify: programming language implementations and database management systems. Programming language implementations (*e.g.,* CPython, Clang, GCC, PHP) have complex codebases with parsers, compilers, garbage collectors, and runtime environments, ideal for testing LLMs on challenging low-level tasks. Database management systems (*e.g.,* MySQL, SQLite, DuckDB) serve as core infrastructure with intricate query optimizers, transaction engines, and storage layers, and their extensive test suites provide rich ground truth for correctness and safety.

Beyond these, OSS-Bench can be flexibly extended to other complex systems, such as web servers and networking stacks (*e.g.,* Nginx, Apache HTTPD), container/orchestration platforms (*e.g.,* Docker, Kubernetes), and ML frameworks (*e.g.,* TensorFlow C-API, PyTorch's C++ frontend). These mature, actively maintained projects offer large codebases and comprehensive test suites, making them excellent live datasets for realistic LLM benchmarking.

**Function Extraction.** OSS-BENCH employs `libclang` (The LLVM Project, 2025), part of the LLVM toolchain, to extract all function definitions including comments from the codebase reliably. Extracted functions are stored in a database and filtered by token count to exclude those that are excessively short or long. In typical large OSS projects, this process yields thousands of functions; by default, we retain functions with 10-256 tokens, filtering out approximately 10% of the corpus.

**OSS-BENCH Coding Prompt.** Our prompt emulates common AI-assisted coding workflows: users provide an existing code snippet and request improvement. As illustrated in Figure 2, it asks the LLM to improve a given code snippet at the function level with two primary goals: *(1) ensure memory safety* and *(2) enhance code efficiency*. The first objective guarantees a fair evaluation of Metric III (memory safety) by explicitly highlighting the attention to code safety. The second objective drives substantive logic improvements, preventing trivial or minimal edits. OSS-BENCH 's prompt framework is also extensible, supporting other tasks like full code synthesis or automated bug repair.

**Metric I — Compilability.** Compilability is one natural and common task in compiled languages. Incompilability indicates syntax (*e.g.,* incorrect grammar) or semantic (*e.g.,* use of undefined variables)

```
                          OSS-Bench Coding Prompt
> You are an expert on C programming.

                  I'm happy to help with any questions you have about C programming, …

> Please optimize the following C function snippet (dont add header files) and (1)
ensure memory safety (2) improve code efficiency. Give the optimized code directly
without explanation. The code is as following:
```=c
static zval *zho_it_get_current_data(zend_object_iterator *iter){
    zend_hooked_object_iterator *hooked_iter = (zend_hooked_object_iterator*)iter;
    zho_it_fetch_current(iter);
    return &hooked_iter->current_data;
}```

                ```=c
                static inline zval *zho_it_get_current_data(zend_object_iterator *iter) {
                    if (!iter) return NULL;
                    zho_it_fetch_current(iter);
                    return &((zend_hooked_object_iterator *)iter)->current_data;
                }```
```

Figure 2: OSS-BENCH uses daily code prompts to simulate common LLM-assisted coding workflows.

errors. OSS-BENCH first compiles the original open-source software to ensure it builds without errors. Then, OSS-BENCH selects each from extracted OSS functions and replaces the original function with its improved version from language models. After code replacement, OSS-BENCH recompiles the entire project and checks its compilability. The compilation result can be success, failure, or unrelated (*i.e.,* changing such functions does not affect the compilation). OSS-BENCH records each function's compilation result, and the overall compilation pass rate (unrelated functions are excluded) is used as the final score.

**Metric II — Functional Testing.** Open-source projects rely on extensive test suites (unit, integration, end-to-end, and regression tests) to validate code updates. To measure this metric, OSS-BENCH first builds the original OSS and runs its official tests to record a baseline. Next, from the set of functions that compiled successfully in Metric I, OSS-BENCH performs 1,000 test iterations: each iteration randomly selects a subset of the compilable functions (*e.g.,* we use $\sim$1% of the total function number, that is, 100 functions for OSS-BENCH$_{php}$ and 73 functions for OSS-BENCH$_{sql}$ per iteration) and replaces the originals to simulate an LLM-driven update. The test suite is then executed, and the pass rate is recorded for each iteration. Finally, we compute the average test-pass rate over all iterations and quantify the degradation relative to the baseline.

**Metric III — Memory Safety.** Memory safety is a fundamental security concern in compiled languages. Bugs such as buffer overflows and double-free errors are known to be harmful and can be maliciously exploited. OSS-BENCH aggregates the sanitizer alerts from compilation and test logs during previous metrics, and reports the number of unique memory safety bugs. A lower final count of sanitizer alerts results in a higher Challenge III score. Inspired by widely deployed OSS fuzzing frameworks such as OSS-Fuzz (Serebryany, 2017), OSS-BENCH extends this metric with an in-depth fuzzing evaluation detailed in Section 3.4. We highlight that the evaluation shows Metric II using test suites is insufficient, as many bugs are observed after the test suite is passed with Metric III.

**Dissimilarity.** Conservative models may game our benchmark by leaving the original function largely unchanged, thereby inflating their scores. To prevent this, we introduce a *Delta* metric that quantifies the average number of differing lines per function, computed using *difflib*, and serves as a configurable impact factor in the final score. Models whose edits fall below a minimum dissimilarity threshold are excluded from evaluation, and higher dissimilarity can enjoy a bonus in the overall score. This ensures that LLMs perform substantive optimizations, in line with our prompt's emphasis on efficiency improvements, rather than relying on conservative or no changes.

**Scoring.** Each model will first be evaluated through metric scores with raw score ranging from 0 to 100 as follows: (i) Raw compilability score ($s_1$): the proportion of valid functions that compile successfully after LLM optimization. Valid functions are those whose modifications could affect the build (*i.e.,* excluding cases where changes do not alter compilation behavior). We obtain $s_1$ by multiplying this proportion by 100, yielding a raw score between 0 and 100. (ii) Raw test score ($s_2$): the average test pass rate of 1,000 iterations of functional testing. We obtain $s_2$ by multiplying this

pass rate by 100, yielding a raw score between 0 and 100. (iii) Raw sanitizer score ($s_3$): begins at 100, with each detected sanitizer alert reducing the score by a certain rate (down to a minimum of 0), so that fewer alerts correspond to a higher score.

OSS-BENCH does not simply sum the three raw scores, as later metrics depend on earlier ones (*i.e.,* functionality is evaluated only for compilable code, and memory safety is assessed only for functional code). We incorporate this dependency in the form of *chained scores* ($c_1, c_2, c_3$) as:

$$c_1 = s_1 \quad \Big| \quad c_2 = s_2 \times \frac{c_1}{100} \quad \Big| \quad c_3 = s_3 \times \frac{c_2}{100}$$

where $s_1$, $s_2$, and $s_3$ are the raw scores for compilation, testing, and sanitization, respectively. For flexible scoring, we introduce weights $w_1, w_2, w_3 \in [0, 1]$, which default to equal values. To discourage trivial edits, we add a *dissimilarity* bonus $d$, defined as the average number of changed lines per function, with a default weight $w_d = 0.1$. The final score $s$ is:

$$s = w_1 \, c_1 + w_2 \, c_2 + w_3 \, c_3 + w_d \, d.$$

**Automation and Scalability.** OSS-Bench is highly automated: no manual test-case design or ground-truth annotation is required. The only human effort needed is to write adapters (*e.g.,* compilation scripts, log parsers) for each open-source software. Scalability is achieved through configurable components, including project selection, prompt templates, sampling rates, metric weights, and even the addition of other new metrics in the OSS development (*e.g.,* static analysis warnings or performance profiles). This flexibility allows OSS-BENCH to scale from single repositories to entire ecosystems, support multiple programming languages, and adapt dynamically as repositories evolve over time.

## 3 EVALUATION

### 3.1 EVALUATION SETUP

**Selected models**. We select 17 models in total as evaluation candidates, including both closed- and open-source variants, across base and instruction-tuned configurations, spanning various parameter sizes and evaluated under the optimal quantization setting (fp16). These models are *GPT-O1* (OpenAI, 2025a), *GPT-O3-Mini* (OpenAI, 2025b), *Claude-3.7-Sonnect* (Anthropic, Inc., 2025c), *Claude-3.5-Haiku* (Anthropic, Inc., 2025b), *Gemini-2.5-Flash* (Google LLC, 2025a), *Llama3.3-70B-Instruct* (Meta Platforms, Inc., 2025a), *CodeLlama-70B-Instruct* (Code Llama, 2025), *Qwen2.5-Coder-32B-Instruct* (Qwen AI, 2025b), *Qwen3-A3B-30B-Instruct* (Qwen AI, 2025c), *Qwen3-8B-Instruct* (Qwen AI, 2025d), *Gemma3-27B-Instruct* (Gemma AI, 2025b), *Qwen2.5-Coder-14B-Instruct* (Qwen AI, 2025a), *DeepSeek-V2-Coder-16B-Instruct* (DeepSeek AI, 2025), *Starcoder2-15B-Instruct* (BigCode Project, 2024), *Phi4-14B* (Phi Labs, 2025), *Mistral-7B-Instruct* (Mistral AI, 2025a), *CodeGemma-7B-Instruct* (Gemma AI, 2025a), sourced from leading organizations such as OpenAI (OpenAI, 2025c), Anthropic (Anthropic, Inc., 2025a), Google (Google LLC, 2025b), DeepSeek (DeepSeek AI, 2023), Alibaba (Alibaba Group, 2025), Meta (Meta Platforms, Inc., 2025b), Microsoft (Microsoft Corporation, 2025), BigCode (BigCode Project, 2025) and Mistral (Mistral AI, 2025b).

**Setup**. We run open-source models using the Ollama (Ollama, Inc., 2025) platform. To ensure reproducibility, we set the random seed to 0 for all models, except for the Claude series, which does not support custom seed settings. OSS-BENCH provides results on 10,534 functions extracted from the PHP interpreter and 7,321 functions from the SQLite3 database engine. OSS-BENCH$_{php}$ is tested in commit *3786cff1f3f3d755f346ade78979976fee92bb48*, OSS-BENCH$_{sql}$ is tested in commit *942c9587698715734715242737dba07ef296b0ef*. We use *pass@k* Kulal et al. (2019); Chen et al. (2021) with $k = 1$ by default to evaluate model performance across the three metrics, given the large number of tasks. Due to limited time and computing resources, we use a subset of models in OSS-BENCH$_{sql}$.

**Task Scale**. We quantify workload in OSS-BENCH$_{php}$ and OSS-BENCH$_{sql}$ along three evaluation axes—*compilability*, *testing*, and *sanitization*. For an OSS project with $N$ functions and $M$ test cases, each model performs $N$ recompilations, $1,000 \times M$ test-suite executions, and $1,000 \times M$ sanitizer-checked runs, for a total of

$$T = N + 2,000 M \quad \text{tasks per model.}$$

Concretely, in OSS-BENCH$_{php}$, $N = 10,534$ and $M \approx 19,000$ yield $T = 38,010,534$ ($\approx 38.0$M) tasks; in OSS-BENCH$_{sql}$, $N = 7,321$ and $M \approx 1,200$ yield $T = 2,407,321$ ($\approx 2.41$M) tasks. The workload is intentionally heavy and standardized: every model receives the same per-function builds and identical repeated test/sanitizer trials, providing a uniform, controlled basis for comparison across models.

Table 1: Overall results (degradation ranking) of OSS-BENCH$_{php}$ and OSS-BENCH$_{sql}$ compared to the baseline open-source projects. Note: * indicates the *instruct* models; thinking is not enabled in all open-source models; all open-source models are under *fp16* quantization.

| Models | Param. Size | Compilability | Func. Test | Mem. Safety | Delta (10%) | Score |
|---|---|---|---|---|---|---|
| **OSS-BENCH$_{php}$— The PHP Interpreter** | | | | | | |
| PHP (baseline) | N/A | 100 | 99.4 | 100 | 0 | 99.6 |
| 🥇 Claude-3.7-Sonnet | N/A | 🥈**92.97** | 🥈**89.61** | 🥈**84.13** | 3.18 | 85.3 |
| 🥈 GPT-O1 | N/A | 🥇**93.45** | 🥉**86.81** | 🥇**89.65** | 2.52 | 85.0 |
| 🥉 GPT-O3-Mini | N/A | 92.72 | 🥇**89.80** | 🥉**84.48** | 2.43 | 84.5 |
| Qwen3-A3B | 30B | 🥉**92.90** | 82.71 | 68.95 | 1.11 | 75.3 |
| Gemini-2.5-Flash | N/A | 88.27 | 76.12 | 59.29 | 1.66 | 66.8 |
| Qwen3 | 8B | 88.36 | 74.93 | 55.84 | 1.52 | 65.4 |
| Claude-3.5-Haiku | N/A | 84.47 | 72.77 | 53.77 | 3.16 | 62.8 |
| Qwen2.5-Coder* | 32B | 83.50 | 68.79 | 38.25 | 2.50 | 56.8 |
| Qwen2.5-Coder* | 14B | 79.08 | 64.41 | 34.11 | 2.10 | 51.2 |
| Llama3.3* | 70B | 70.89 | 62.17 | 47.56 | 3.40 | 48.7 |
| Gemma3* | 27B | 71.32 | 63.28 | 30.66 | 1.96 | 45.4 |
| StarCoder2* | 15B | 73.35 | 56.81 | 12.37 | 2.54 | 42.6 |
| Phi4 | 14B | 66.59 | 58.53 | 25.14 | 3.12 | 41.6 |
| DeepSeek-Coder-V2* | 16B | 67.06 | 44.17 | 23.76 | 2.09 | 36.7 |
| CodeGemma* | 7B | 65.48 | 41.76 | 12.37 | 1.55 | 33.6 |
| CodeLlama* | 70B | 53.52 | 34.53 | 18.93 | 2.67 | 27.8 |
| Mistral* | 7B | 41.26 | 28.26 | 0.01 | 2.87 | 20.5 |
| **OSS-BENCH$_{sql}$— SQLite3 Database Engine** | | | | | | |
| SQLite (baseline) | N/A | 100 | 100 | 100 | 0 | 100 |
| 🥇 GPT-O3-Mini | N/A | 🥇**96.68** | 🥇**91.81** | 🥇**72.16** | 2.38 | 85.5 |
| 🥈 Qwen3-A3B | 30B | 🥈**96.34** | 🥈**72.03** | 🥈**55.92** | 1.50 | 69.7 |
| 🥉 Qwen3 | 8B | 🥉**94.80** | 🥉**69.83** | 🥉**49.83** | 1.59 | 66.3 |
| Claude-3.5-Haiku | N/A | 92.59 | 60.65 | 37.50 | 2.61 | 59.2 |
| Qwen2.5-Coder* | 32B | 92.47 | 55.04 | 33.59 | 1.85 | 55.3 |
| Qwen2.5-Coder* | 14B | 90.31 | 49.58 | 35.04 | 1.77 | 52.0 |
| Phi4 | 14B | 78.71 | 39.31 | 35.91 | 2.91 | 43.2 |
| Deepseek-Coder-V2* | 16B | 82.74 | 45.46 | 7.20 | 1.98 | 43.0 |
| StarCoder2* | 15B | 81.61 | 41.33 | 0.01 | 1.91 | 40.4 |

**Experimental Infrastructure.** Experiments were conducted across multiple hardware configurations: CPUs (AMD EPYC 9184X, Intel Core Ultra 9 285K, AMD Ryzen 9 9950X), GPUs (NVIDIA A40, H100), and system memory ranging from 32 GB to 512 GB. All experiments were performed on Ubuntu 22.04 and Ubuntu 24.04.

### 3.2 EFFECTIVENESS OF OSS-BENCH

We show our benchmarks, OSS-BENCH$_{php}$ and OSS-BENCH$_{sql}$, can effectively profile LLM coding capabilities and uncover insightful observations through evaluating three metrics proposed in OSS-BENCH.

**Overall Results.** Table 1 summarizes the evaluation of OSS-BENCH$_{php}$ and OSS-BENCH$_{sql}$, reporting raw scores $s_1$, $s_2$, and $s_3$ for compilability, testing, and sanitization, respectively, along with the dissimilarity bonus $d$ and the final chained score $s$. The penalty for each unique sanitizer alert is configurable; by default, we deduct 0.69 points per alert in OSS-BENCH$_{php}$ and 0.19 points in OSS-BENCH$_{sql}$, while memory-leak alerts incur only half the standard deduction. Scores are rounded to two decimal places. The original OSS projects (PHP and SQLite3) serve as baselines: all raw scores are 100 after we evaluate the same metrics, except PHP's test suite, which yields 99.4 due to some expected failing tests. Models that incur less degradation relative to these baselines achieve higher final scores, indicating stronger real-world coding capabilities.

Examining OSS-BENCH$_{php}$ reveals a clear performance hierarchy. At the top, closed-source, instruction-tuned models consistently achieve the highest compilation and testing scores while generating the fewest sanitizer alerts. A notable outlier is GPT-O3-Mini, which, despite its smaller size, matches or exceeds the performance of larger mid-tier models across all three metrics. Moving down the ranks, mid-tier models begin to show significant degradation in memory safety, even when their compilation and test-pass rates remain respectable. Finally, smaller open-source models exhibit the steepest declines, particularly in triggering memory-safety issues, highlighting that neither parameter count nor general coding ability alone guarantees robust security in complex, real-world codebases.

**Qwen Family Comparison.** Analyzing the Qwen family on OSS-BENCH$_{php}$ and OSS-BENCH$_{sql}$ reveals clear performance gains across generations. The third-generation Qwen3 models, especially Qwen3-A3B-30B,

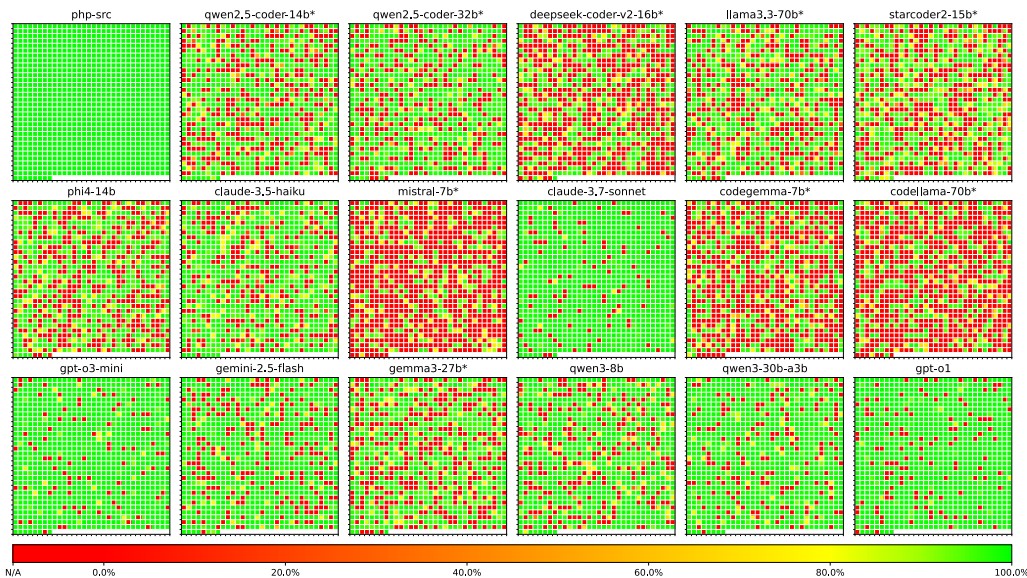

Figure 3: Visualized Test Pass Rates of 1,000 Test Iterations in OSS-BENCH$_{php}$

consistently outperform their Qwen2.5 predecessors in all three metrics (compilability, functional testing, and memory safety). For example, Qwen3-A3B-30B achieves higher compilation success and test-pass rates than Qwen2.5-Coder-32B, while also triggering fewer sanitizer alerts. Remarkably, even the smallest Qwen3-8B model surpasses the much larger Qwen2.5-Coder-32B across every metric, demonstrating that architectural improvements outweigh mere parameter count. Notably, however, the Qwen3 series tends to make more conservative edits, reflected in lower dissimilarity scores, indicating that these models achieve stronger performance through targeted code modifications rather than extensive rewrites.

**Model Size.** Model size does not reliably predict performance. For instance, on OSS-BENCH$_{php}$ the 70B-parameter CodeLlama scores lower across all three metrics than several lighter models (7B–32B). Likewise, in OSS-BENCH$_{sql}$ the 8B-parameter Qwen3-8B outperforms most other open-source models, highlighting that architecture and training strategy matter more than sheer parameter count in real-world coding tasks.

**Test Result Analysis.** Figure 3 visualizes the results of 1,000 functional test iterations for OSS-BENCH$_{php}$, covering the PHP baseline and 17 LLMs. Each dot represents the average test pass rate for a single iteration, color-coded from red (0%) to green (100%). Deep red indicates N/A, corresponding to cases where the test suite fails to start after incorporating LLM-generated code. The visualization clearly differentiates model performance: top performers (*i.e.,* GPT-O1, GPT-O3-Mini, and Claude-3.7-Sonnet) exhibit predominantly green dots and few failures, whereas models such as Mistral-7B-Instruct, CodeGemma-7B-Instruct, and CodeLlama-70B-Instruct display numerous red and yellow dots, reflecting frequent test failures or low pass rates.

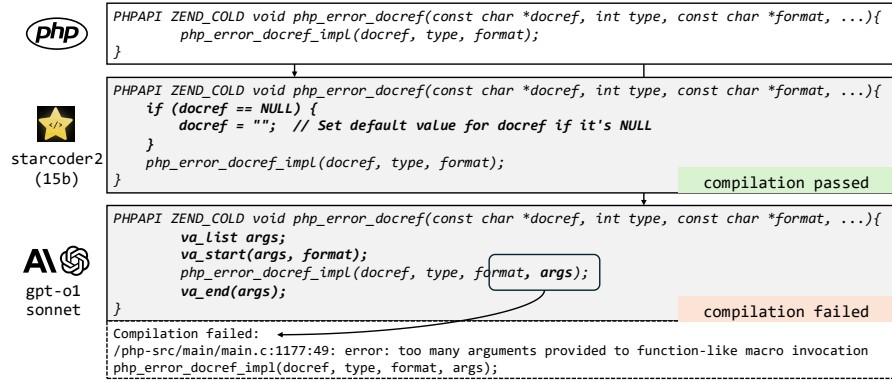

Figure 4: More memorization adds incorrect statements in advanced models

Figure 5: Models need caution when generating code for familiar content, such cases may not require optimization, but the presence of elements like *lxb_encoding* and *iso-8859-2* thills LLMs.

## 3.3 FAIRNESS OF OSS-BENCH

We demonstrate that OSS-BENCH can fairly assess LLM coding capabilities even under knowledge contamination. First, we probe each model's ability to identify OSS function-level snippets and analyze the link between memorization and coding performance. Then, through two case studies, we show that overfitting to these challenging tasks, driven by excessive memorization, can produce incorrect edits in high-end models, whereas lighter models remain robust.

We use a contamination prompt to probe model memorization of OSS source code (details and examples in the Appendix). Our results show that LLMs readily identify a project when explicit identifiers, such as the PHP data structure *zval*, are present. When identifiers are removed, only top-tier models (*e.g.,* GPT-O1, Claude-3.7-Sonnet) occasionally recognize the source, while most models fail on isolated snippets. We observe that higher memorization does not equate to better coding performance: Qwen3-30B-A3B, which recognizes only snippets with obvious keywords, outperforms Gemini-2.5-Flash and Claude-3.5-Haiku (both of which infer sources without PHP-specific identifiers) across all metrics, despite its lower memorization.

More memorization can sometimes be counterproductive. Figure 4 shows a compilation error produced by some top models (Table 1), while others compile the same variadic function in main.c without issue. In their optimized versions, these models injected extra error-handling statements and passed an unintended args parameter, leading to "error: too many arguments provided." Our analysis of the PHP codebase reveals that the injected statements mirror common error-handling patterns in PHP, explaining why GPT-O1 and Claude-3.7-Sonnet, drawing on their extensive training, introduced them despite breaking this particular function call. Thus, OSS-BENCH mitigates overfitting by leveraging large, complex OSS codebases where memorizing every detail is impractical and can lead to internal inconsistencies.

Figure 5 illustrates one another compilation case in OSS-BENCH$_{php}$ where only Qwen3-30B-A3B successfully compiles a simple encoding function for *iso-8859-2*. Despite the function's minimal logic that requires no optimization, other models (*e.g.,* GPT-O1) inject unrelated statements when prompted to improve memory safety and efficiency, causing compilation failures. Qwen3-30B-A3B's restraint highlights its ability to recognize when no change is needed, avoiding unnecessary edits that break the build. This case highlights that, although language models can memorize extensive patterns, they must exercise restraint in familiar contexts and rethink whether additional modifications are truly necessary according to the user prompt.

We highlight that OSS-BENCH advantage on overfitting mitigation leverages the rapidly evolving nature of open-source software. For example, the PHP project undergoes daily code commits in its actively maintained repository. In such a dynamic environment, overfitting to a static knowledge base becomes impractical, as model memorization can quickly become outdated, potentially leading to incorrect predictions or unexpected failures. Another contributing factor is the use of random sampling in our approach, which generates millions of possible task combinations, making it difficult for models to memorize the exact sampled benchmarks.

## 3.4 EXTENDED MEMORY-SAFETY EVALUATION VIA FUZZING

In Section 2, OSS-BENCH measures memory safety by enabling sanitizers at compile time and extracting alerts from compilation and test logs. This approach is efficient, reproducible, and sufficient to distinguish model behaviors. However, compilation and standard tests only cover a limited set of program states, leaving many potential memory bugs undiscovered. Fuzz testing (*e.g.,* randomized input generation) is a well-established method for uncovering deep bugs.

To extend our evaluation, we selected the top-performing closed- and open-source models (GPT-O1 and Qwen3-A3B-30B) along with the PHP baseline and fuzzed the first 500 valid sampled test iterations from Metric II. Using the state-of-the-art PHP fuzzer FlowFusion (Jiang et al., 2025), we generated 50,000 mutated inputs per iteration and recorded the unique sanitizer alerts. This extended analysis provides a more rigorous comparison of each model's memory-safety capabilities.

| Models | Param. Size | #SEGV | #Spatial | #Temporal | #Undefined | Total |
|---|---|---|---|---|---|---|
| PHP (baseline) | N/A | 3 | 37 | 1 | 15 | 56 |
| GPT-O1 (OpenAI, 2025a) | N/A | 101 | 64 | 8 | 163 | 336 |
| Qwen3-A3B (Qwen AI, 2025c) | 30B | 211 | 164 | 11 | 350 | 736 |

The table above presents deduplicated memory safety bug counts via our extended fuzzing. We categorize bugs into four types: *SEGV* (*i.e.,* segmentation faults), *Spatial* (*e.g.,* stack or heap overflows), *Temporal* (*e.g.,* use-after-free or double-free), and *Undefined* (other C/C++ undefined behavior, *e.g.,* integer overflows). After 500 fuzzing iterations, even the original PHP code shows exhibits some safety bugs, establishing a realistic baseline. Despite prompts emphasizing memory safety, LLM-generated edits introduce roughly ten times more new violations. Among the evaluated models, GPT-O1 (closed) and Qwen3-A3B-30B (open) produce the fewest new bugs in Table 1; all other models generate substantially more. These findings underscore the significant room for improvement in LLMs' handling of low-level memory safety.

## 3.5 DISCUSSION

**Prompt Flexibility.** We focus exclusively on realistic code-editing prompts for OSS-BENCH$_{php}$ and OSS-BENCH$_{sql}$ as it is computationally expensive to experiment with various prompts. However, the prompt in OSS-BENCH is flexible and can be extended to more aggressive tasks, such as direct code synthesis or structural transformations (*e.g.,* altering control or data flow).

**Scoped Scalability.** Although the compilation challenge can be extended to any compiled language, such as C, C++, Rust, Go, Swift, and Java, the sanitizer checks in Metric III are currently limited to memory-safety issues in C/C++ and Rust. While this focus offers a novel dimension for assessing code quality, it constrains the applicability of the sanitizer challenge to other languages and safety domains.

**Time-Cost Evaluation.** OSS-BENCH typically takes longer evaluation than other benchmarks, driven by the large number of functions in each OSS project and the extensive testing required for robust results. On a typical PC, the evaluation for each model completes in approximately 48 hours for *pass@1*. We consider this time investment reasonable because the evaluation cycle is much shorter than the typical month-level intervals between major LLM updates.

## 4 RELATED WORK

Recent benchmarks span realistic production and contest-based tasks (BigCodeBench (Zhuo et al., 2024), LiveCodeBench (Jain et al., 2024)), multilingual and class-level generation (HumanEval-X (Zheng et al., 2023), HumanEval-XL (Peng et al., 2024), ClassEval (Du et al., 2023)), practical workflows and library integration (SWE-Bench (Jimenez et al., 2023b), ML-Bench (Tang et al., 2023)), domain-specific and instruction-compliant scenarios (DomainCodeBench (Zheng et al., 2024a), CodeIF (Wang et al., 2025)), evolution-aware and complexity-controlled generalization (HumanEvo (Zheng et al., 2025), DynaCode (Hu et al., 2025), Python-Saga (Yadav & Singh, 2024), ComplexCodeEval (Feng et al., 2024)), security-focused evaluations (CodeLM-Sec (Hajipour et al., 2024), SecRepoBench (Dilgren et al., 2025), CWEval (Peng et al., 2025)), and holistic quality and competitive ranking (RACE (Zheng et al., 2024b), CodeArena (Baars & Meester, 2019), CodeElo (Quan et al., 2025)). Further discussion can be found in the Appendix.

## 5 CONCLUSION

We present OSS-BENCH, an automatic benchmark generator that constructs live, large-scale evaluation tasks from real-world open-source software by reusing novel ground truth signals (*i.e.,* compilation results, functional test outcomes, and sanitizer checks). Instantiated on the PHP interpreter and SQLite3 engine, OSS-BENCH$_{php}$ and OSS-BENCH$_{sql}$ can effectively profile the coding capabilities of large language models and reveal insightful patterns, such as intra-family behavior and inconsistencies between model size and performance. Furthermore, OSS-BENCH mitigates overfitting by leveraging the rapidly evolving nature of open-source projects and employing random sampling over dynamic datasets. The framework is scalable across project selection, prompt templates, sampling rates, and metric weights. Future work will extend OSS-BENCH to additional compiled languages like Rust, incorporate richer code context, introduce new metrics such as efficiency profiling, and optimize the fuzzing pipeline to support continued progress in AI-assisted coding tools.

ETHICS STATEMENT

We evaluate LLM-driven code edits on publicly available OSS under original licenses. No human subjects or personally identifiable data are involved. We document compute settings and software configurations to ensure reproducibility, and design OSS-BENCH to enable rigorous yet low-burden evaluation for OSS communities.

OPEN SCIENCE STATEMENT

We are committed to openly sharing all research artifacts associated with this work. We upload our source code to the supplementary material. Our commitment to open science aligns with the broader initiative to foster transparency and collaboration within the research community.

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

APPENDIX

## 5.1 DETAILED RELATED WORK

Recent benchmarks have increasingly emphasized realistic and dynamic tasks to evaluate the coding capabilities of large language models (LLMs). BigCodeBench (Zhuo et al., 2024) assembles realistic programming challenges involving diverse APIs and unit tests to simulate production-level coding, while LiveCodeBench (Jain et al., 2024) sources tasks from recent coding contests to mitigate data leakage and reduce overfitting risks. HumanEval-X (Zheng et al., 2023) and HumanEval-XL (Peng et al., 2024) extend multilingual and cross-lingual code generation capabilities across multiple programming and natural languages, whereas ClassEval (Du et al., 2023) evaluates comprehensive class-level Python code generation tasks.

Benchmarks such as SWE-Bench (Jimenez et al., 2023b;a) and ML-Bench (Tang et al., 2023) focus on practical software engineering workflows, targeting real-world GitHub issues, multi-file bug fixes, and integrations with popular machine learning libraries. DomainCodeBench (Zheng et al., 2024a) and CodeIF (Wang et al., 2025) expand the evaluation scope to domain-specific scenarios and instruction-following tasks, assessing models' adaptability across diverse software ecosystems and adherence to detailed human instructions.

To test generalization and adaptability over software evolution, HumanEvo (Zheng et al., 2025) prompts LLMs with outdated codebase snapshots, requiring forward-compatible code modifications, while DynaCode (Hu et al., 2025) programmatically generates tasks of varying complexity to evaluate true algorithmic reasoning. PythonSaga (Yadav & Singh, 2024) similarly focuses on generalization, examining programming concepts across varying difficulty levels. ComplexCodeEval (Feng et al., 2024) assesses multifaceted tasks from large Java and Python repositories to evaluate broader software development contexts comprehensively.

Security-focused benchmarks, including CodeLMSec (Hajipour et al., 2024), SecRepoBench (Dilgren et al., 2025), and CWEval (Peng et al., 2025), rigorously evaluate models' secure coding capabilities through vulnerability-aware code generation and static analysis. Holistic quality assessments are represented by RACE (Zheng et al., 2024b), which measures code correctness, readability, maintainability, and efficiency, and CodeArena (Baars & Meester, 2019), which employs adaptive scoring based on collective model performance. Finally, CodeElo (Quan et al., 2025) ranks models on competitive programming tasks using Elo ratings, providing human-comparable skill evaluations.

## 5.2 EVALUATION RESULT ANALYSIS ON COMPILABILITY OF OSS-BENCH$_{php}$ AND OSS-BENCH$_{sql}$

We demonstrate the results in Figure 6 and Figure 7. Each panel visualizes *500 independent function-level edits*; the horizontal axis is function ID and *each dot is a compilation failure* for that function under a given model (lower dot density is better). This is the first, fully automated gate in OSS-BENCH: after a model rewrites one C/C++ function (PHP or SQLite), we rebuild the project; any compiler error (*e.g.,* unknown identifier, wrong type/arity, missing include/macro, misuse of internal API) is counted as a failure. The baseline projects (*php-src* and *sqlite*) appear with empty panels, confirming that all dots represent regressions introduced by the model edits.

In OSS-BENCH$_{php}$, the ranking by compile score shows a clear separation: frontier instruction-tuned models exhibit few failures and relatively uniform scatter, while several open-source code models accumulate many failures with conspicuous "hot zones" where multiple models break on the same function ranges. Such clustering indicates shared brittleness around PHP internals (*e.g.,* refcounted *zval* handling, engine macros, extension-specific headers), not random syntax slips. Qualitatively, the best three performers (*e.g., gpt-o1, claude-3.7-sonnet, qwen3-30b-a3b*) keep failure dots sparse across the 0–500 window, while other models show

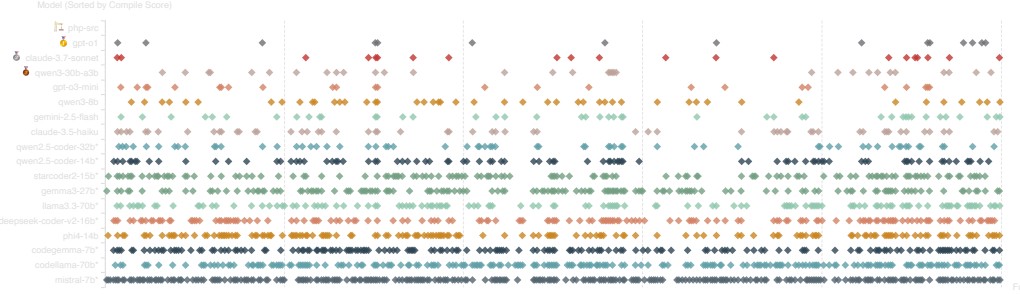

Figure 6: Compilability Results in OSS-BENCH$_{php}$

bands of dense dots, typical of missing symbol wiring or incompatible call signatures inside Zend/extension APIs.

In OSS-BENCH$_{sql}$, we observe the same overall pattern but on a codebase with heavier macro use and stricter invariants: the top models (*e.g., gpt-o3-mini, qwen3-30b-a3b, qwen3-8b*) produce the fewest compilation breaks, whereas mid-tier code models exhibit clusters of failures around functions that touch VDBE op emission, B-Tree/pager interfaces, build-time feature macros, or error-propagation paths. Again, dots align across models for particular function IDs, signaling that difficult, infrastructure-heavy internals (rather than idiosyncratic mistakes) are the primary source of build breakage.

**Takeaways.** (1) Compilability cleanly stratifies models before testing or sanitizers are considered; panels with few, evenly spread dots correlate with stronger downstream performance. (2) Failure *clusters* reveal "API gravity wells" where edits require precise header context, macros, and calling conventions, natural pain points for LLMs without oracle hints. (3) Because the metric is fully automated and tied to real build systems, improving fixtures (*e.g.,* supplying minimal context headers, consistent include paths, or read-only API stubs) can reduce false starts without relaxing the benchmark's no-human, no-LLM-oracle principle.

### 5.3 EVALUATION ON MODULE ANALYSIS OF OSS-BENCH$_{php}$

Figure 8 shows *module-specific* pass rates for the **Functional Test** metric across PHP extensions. Because OSS-BENCH relies exclusively on native project signals (PHPT tests), the scores reflect how often model edits preserve or improve observable behavior *without* bespoke harnessing or external oracles. We observe three bands:

1. **Self-contained, deterministic modules (near-saturated).** Parsing/compute–centric extensions (*e.g., op-cache, phar, pcre, mbstring, reflection, json, xmlreader/xmlwriter/xml, zlib, sodium, spl, sqlite3*) yield ≈98–100% with minimal spread across models, indicating that when the tests have few environmental preconditions, top models behave similarly and reliably.

2. **OS/Environment-interactive modules (moderate headroom).** Extensions such as *libxml, sockets, pcntl, shmop,* and *readline* land in the ∼57–80% band. Their tests are sensitive to TTY, process control, or shared-memory behavior; models cluster but small gaps appear, suggesting robustness differences rather than missing semantics.

3. **Service-/configuration-backed modules (uniformly difficult).** Database and network stacks (*mysqli, pdo_mysql, pgsql, pdo_pgsql, snmp*) and configuration-heavy modules (*ldap, gettext*) show low pass rates (single-digit to ∼50%). Under OSS-BENCH's non-intervention policy, tests that assume live services, credentials, or locale setup remain challenging for *all* models, reflecting infrastructure requirements rather than oracle design or manual tuning.

Because the benchmark derives ground truth strictly from the OSS pipeline, lifting the lower band does not require new labels or LLM judges; it requires *better fixtures*. Practical avenues include ephemeral MySQL/PostgreSQL/LDAP/SNMP containers with seeded schemas/creds, locale provisioning for *gettext*, and controlled privilege settings for *pcntl*/shared-memory tests. Such changes increase task realism while preserving the benchmark's core principle: *no human or LLM in the oracle loop*.

### 5.4 MEMORY SAFETY ANALYSIS

We visualize the memory safety results in Figure 9 and Figure 10.

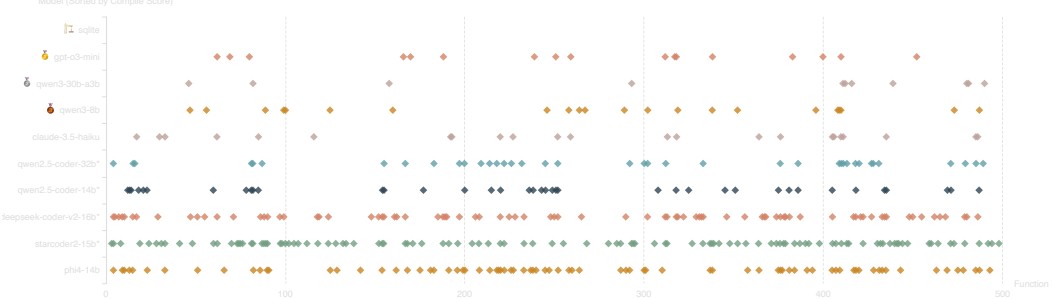

Figure 7: Compilability Results in OSS-BENCH$_{sql}$

**Sanitizer (memory-safety) metric.** For each *function-level* edit a model proposes to the PHP interpreter (C/C++), OSS-BENCH$_{php}$ rebuilds with AddressSanitizer (ASan) and runs the upstream PHPT test suites; any ASan report (*e.g.,* out-of-bounds access, use-after-free, double free) is recorded as a memory-safety alert. This ground truth is fully automated, no human judges or LLM oracles, and the clean *php-src* baseline yields **0** alerts, so all counted issues are regressions introduced by model edits.

**Visualization and counting.** In the figure, each dot denotes one *unique* alert after deduplication by a normalized backtrace signature (error kind + top frames + source path), keyed to the specific edit; lower dots-per-panel is better. Across models we observe roughly **16–170** alerts per model, with frontier systems tending toward the low end and several open models producing dozens to low hundreds; numbers are orthogonal to functional pass rates (a patch can pass tests yet still be unsafe).

**Why it matters.** The sanitizer metric complements *Compilability* and *Functional Test* by surfacing low-level defects invisible to black-box testing, providing a precise, scalable signal aligned with real engineering practice. It highlights common failure modes (predominantly OOB and UAF) in native PHP components and allows continuous, hands-off evaluation as upstream code evolves.

## 5.5 CONTAMINATION PROMPT AND EXAMPLE FUNCTION.

The system prompt is *"You are an expert on open-source software"*. The contamination test prompt is formulated as follows: *"Could you recognize the following C function snippet and provide the name of the open-source software it belongs to using the format 'Yes. It belongs to ...'? If you are unsure, please respond with 'I dont know'. The code snippet is: {php function}"*. If the answer contains "php", this model is believed to have contaminated memory of the PHP project.

We assess potential memorization by testing LLMs on selected functions from OSS-BENCH$_{php}$ that omit explicit identifiers such as "PHP" or "Zend". We demonstrate some function examples as follows:

Listing 1: The PHP function (difficult) that is recognized by no models

```
void KeccakP1600_ExtractBytesInLane(const void *state, unsigned int lanePosition,
unsigned char *data, unsigned int offset, unsigned int length)
{
    UINT64 lane = ((UINT64*)state)[lanePosition];
#ifdef KeccakP1600_useLaneComplementing
    if ((lanePosition == 1) || (lanePosition == 2) || (lanePosition == 8) ||
    (lanePosition == 12) || (lanePosition == 17) || (lanePosition == 20))
        lane = ~lane;
#endif
#if (PLATFORM_BYTE_ORDER == IS_LITTLE_ENDIAN)
    {
        UINT64 lane1[1];
        lane1[0] = lane;
        memcpy(data, (UINT8*)lane1+offset, length);
    }
#else
    unsigned int i;
    lane >>= offset*8;
    for(i=0; i<length; i++) {
        data[i] = lane & 0xFF;
        lane >>= 8;
    }
#endif
}
```

Listing 2: The PHP function (medium) that is only recognized by GPT-O1 and Claude-3.7-Sonnet

```
static void gc_scan_roots(gc_stack *stack)
{
        uint32_t idx, end;
        gc_root_buffer *current;
        /* Root buffer might be reallocated during gc_scan,
         * make sure to reload pointers. */
        idx = GC_FIRST_ROOT;
        end = GC_G(first_unused);
        while (idx != end) {
                current = GC_IDX2PTR(idx);
                if (GC_IS_ROOT(current->ref)) {
                        if (GC_REF_CHECK_COLOR(current->ref, GC_GREY)) {
                                GC_REF_SET_COLOR(current->ref, GC_WHITE);
                                gc_scan(current->ref, stack);
                        }
                }
                idx++;
        }
        /* Scan extra roots added during gc_scan */
        while (idx != GC_G(first_unused)) {
                current = GC_IDX2PTR(idx);
                if (GC_IS_ROOT(current->ref)) {
                        if (GC_REF_CHECK_COLOR(current->ref, GC_GREY)) {
                                GC_REF_SET_COLOR(current->ref, GC_WHITE);
```

```
25                                    gc_scan(current->ref, stack);
26                            }
27                    }
28                    idx++;
29            }
30 }
```

Listing 3: The PHP function (easy) that is recognized by all models (due to the use of "zval")

```
1 static void
2 describe_dict_fn (const char * const lang,
3                   const char * const name,
4                   const char * const desc,
5                   const char * const file,
6                   void * ud) /* {{{ */
7 {
8        zval *zdesc = (zval *) ud;
9        array_init(zdesc);
10       add_assoc_string(zdesc, "lang", (char *)lang);
11       add_assoc_string(zdesc, "name", (char *)name);
12       add_assoc_string(zdesc, "desc", (char *)desc);
13       add_assoc_string(zdesc, "file", (char *)file);
14 }
```

## 5.6  OSS-BENCH$_{sql}$ TEST RESULTS

We present the OSS-BENCH$_{sql}$ test results in the Figure 11.

## 5.7  THE USE OF LARGE LANGUAGE MODELS (LLMS)

This paper uses LLMs as a general-purpose assist tool for polishing text.

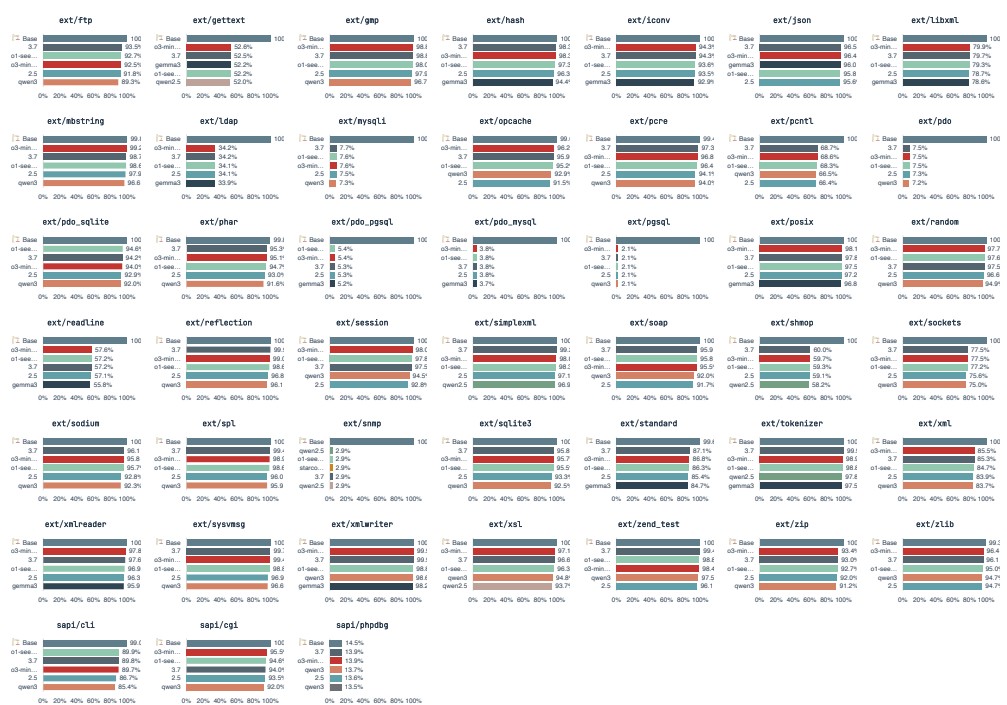

Figure 8: **Module-specific pass rates on OSS-Bench (PHP) for the top five models.** Each small multiple shows one PHP module and the bar-wise pass rate (%) of the five best-performing LLMs on that module's tests. Modules are arranged in a grid for quick, side-by-side comparison. Higher is better.

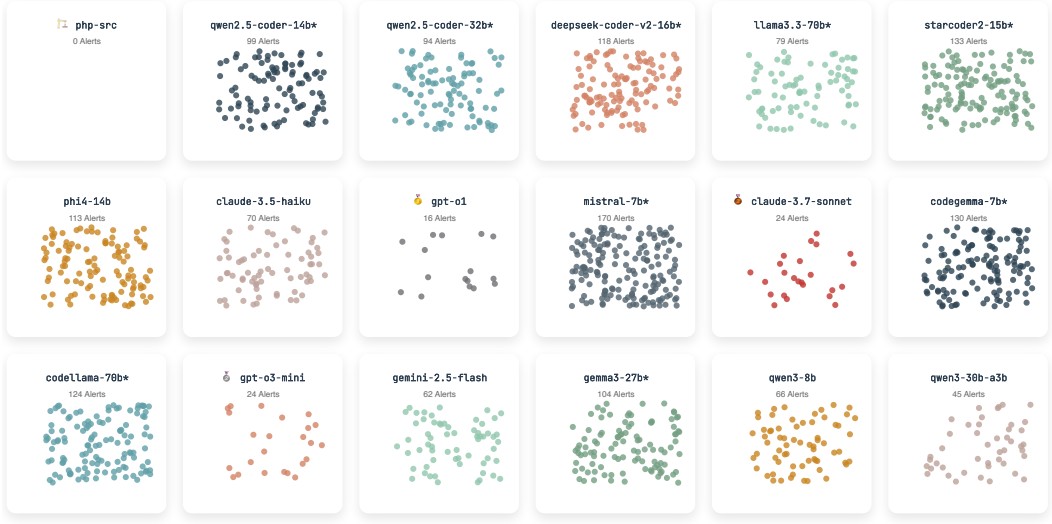

Figure 9: **OSS-BENCH$_{php}$ Sanitizer Score Visualization.** Each dot is one *unique* sanitizer alert triggered after replacing a single PHP C/C++ function with the model's edit and running the upstream PHPT test suites. Lower is better. The baseline (*php-src*) has 0 alerts.

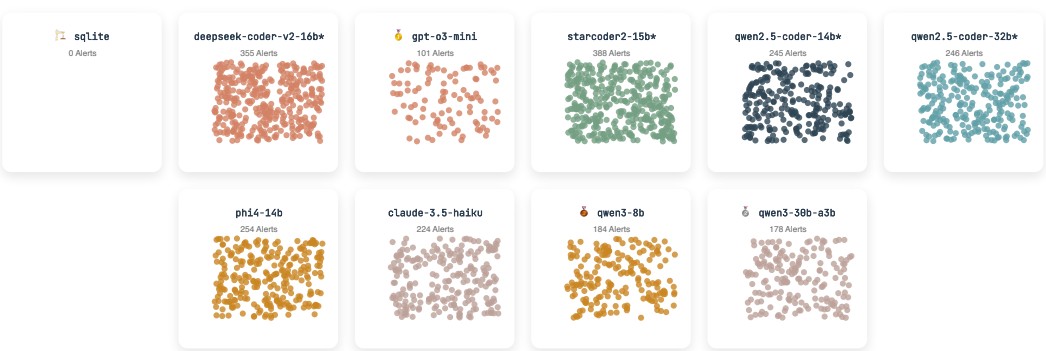

Figure 10: **OSS-BENCH$_{sql}$ Sanitizer Score Visualization.** Each dot is one *unique* sanitizer alert triggered after replacing a single SQLite3 C/C++ function with the model's edit and running the upstream SQLite3 test suites. Lower is better. The baseline (*sqlite3*) has 0 alerts.

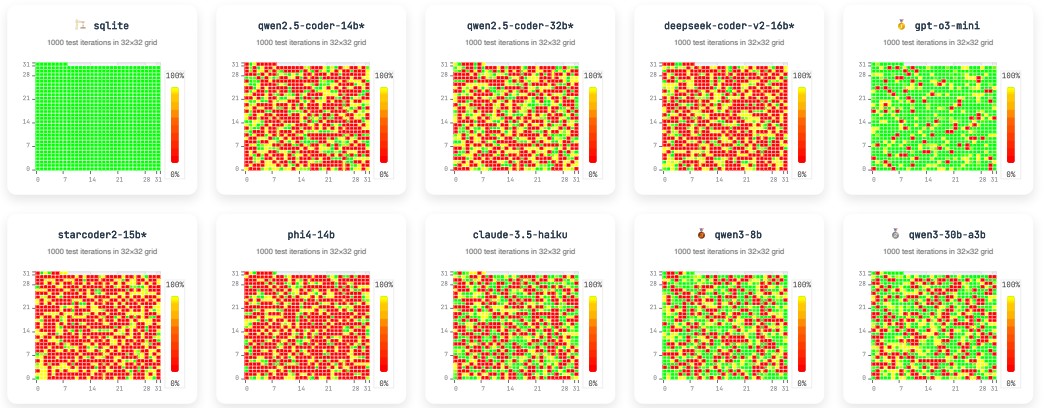

Figure 11: Visualized Test Pass Rates of 1,000 Test Iterations in OSS-BENCH$_{sql}$

