# OpenReview forum: "OSS-Bench: Evaluating LLMs via the Realistic Open-Source Software Development Pipeline"
_ICLR.cc/2026/Conference — Submitted to ICLR 2026_

### Official Review · Reviewer_kpz1 · 2025-10-22

**Soundness:** 3
**Presentation:** 3
**Contribution:** 2
**Rating:** 4
**Confidence:** 3

**Summary:**

The paper presents OSS-BENCH, an automated benchmarking framework designed to evaluate LLMs’ code editing capabilities across three key dimensions: compilability, functional correctness, and memory safety. It aims to overcome the limitations of existing benchmarks, such as Manual effort, less challenging tasks and limited focus on code security.

**Strengths:**

The paper provides a comprehensive evaluation of 17 LLM models on code-editing tasks, offering valuable insights into their comparative performance.

**Weaknesses:**

For the OSS-Bench, lack deep analysis and comparison with previous benchmark to clarify the dataset address limitations like less challenging, limited focus on code security.

The benchmark construction process appears overly straightforward: function extraction from projects, test cases execution, and applying sanitizer tools for memory safety evaluation. While simplicity is not inherently a weakness, the paper should better articulate the novelty and practical value of OSS-BENCH beyond serving as a basic automated pipeline. From this perspective, it is difficult to consider OSS-BENCH a true benchmark rather than a simple workflow.

Moreover, since nearly all functions are included in the evaluation, it remains unclear how this design advances future research or benefits LLMs-based improvement. The current setup relies on only two projects (PHP and SQLite3) with more than 10,000 functions each. Such a narrow yet uniform treatment of all functions makes the approach unrealistic to scale or generalize to more projects in practice.

**Questions:**

Why were only two projects—the PHP interpreter and SQLite3 database engine—selected for evaluation? Since only two projects are included, how do the authors address potential dataset bias and ensure the generalizability and representativeness of the evaluation results?

What advantages do these two projects offer compared with those used in previous studies? Can the authors provide quantitative evidence, such as function sampling or statistical comparisons, to support their selection? Only say “they are complex seems subjective”.

The function extraction process appears to include almost all functions for evaluation. Since the paper’s contribution focuses on the benchmark, are all these functions needed to assess LLMs’ code-editing capabilities, particularly in improving memory safety and efficiency? Given that the best results for Compilability, Functional Test, and Memory Safety exceed 90%, 90%, and 70% respectively, does this imply that current LLMs already demonstrate strong code-editing ability? If so, the paper might consider releasing or analyzing a smaller, representative subset to facilitate further research and fair comparison among LLMs.

It would be valuable to analyze why LLMs fail on certain functions—what types of code or security aspects remain challenging? Providing insights into these failure cases could substantially strengthen the contribution and deepen our understanding of current LLM limitations.

---

### Official Review · Reviewer_iSmM · 2025-10-23

**Soundness:** 3
**Presentation:** 3
**Contribution:** 3
**Rating:** 4
**Confidence:** 4

**Summary:**

This paper addresses the rapid rise of AI coding assistants like Copilot and Cursor, highlighting the need for robust evaluation of LLM-generated code quality. Existing benchmarks suffer from issues such as heavy manual effort for static datasets, indirect or unchallenging tasks, non-scalable ground truth, and neglect of low-level security (e.g., memory safety). The authors introduce OSS-BENCH, an automated benchmark generator that creates large-scale, dynamic evaluation tasks from real-world Open-Source Software (OSS). It replaces functions with LLM-generated code and evaluates using three novel metrics: compilability (success rate via compilation failures), functional correctness (test suite pass rate degradation), and memory safety (sanitizer alerts for issues like buffer overflows).

**Strengths:**

- No manual labeling or LLM-based ground truth; leverages OSS's natural signals (compilation/tests/sanitizers) for dynamic updates as repositories evolve, reducing overfitting to static data.

- Draws from mature OSS (e.g., PHP, SQLite3) with complex low-level code (e.g., garbage collectors, query optimizers), mirroring actual development pipelines in compiled languages—more demanding than simple algorithms.

- Innovates with memory safety (sanitizers + fuzzing), addressing a gap in prior benchmarks; chained scoring accounts for metric dependencies, and Delta encourages meaningful optimizations.

- Million-scale evaluation uncovers patterns like intra-family behaviors (Qwen progress), size-performance inconsistencies, and test suites' inadequacy for security.

**Weaknesses:**

- The paper claims that OSS-BENCH is “the first fully automated OSS pipeline without human or LLM intervention,” yet similar efforts from 2023–2025 diminish this uniqueness. SWE-bench pioneered automated evaluation of LLM-based bug fixing using GitHub OSS issues, covering compilation and testing at a million-task scale to mitigate overfitting. SecRepoBench focuses on secure code generation in real repositories, employing sanitizers to assess memory vulnerabilities—overlapping substantially with OSS-BENCH’s three metrics. BUILD-BENCH  targets OSS compilation agents with dynamic function replacement and superior resource efficiency. SEC-bench and SecureAgentBench extend to agent-level security tasks, including fuzzing. Even earlier work like HumanEval+ leveraged unit-test ground truth. These precedents make OSS-BENCH’s “novel metrics” and “live updates” appear incremental rather than revolutionary.

- The paper introduces memory safety innovations (sanitizer + fuzzing, inspired by OSS-Fuzz), but these are limited to low-level issues such as buffer overflows and use-after-free, while neglecting higher-level vulnerabilities like SQL injection in OSS-BENCHsql, concurrency deadlocks, and side-channel attacks. Although the authors acknowledge insufficient testing (bugs exposed post-fuzzing), they fail to quantify fuzzing costs or integrate static analysis tools (e.g., Coverity). The degradation-focused evaluation overlooks positive improvements such as efficiency benchmarks or runtime profiling, and the weighting mechanism remains subjective and unvalidated. Dynamic updates rely on OSS commits, yet upstream changes like security patches may break consistency due to the absence of version pinning. Compared to SEC-bench, which integrates PoC generation and multi-dimensional security tasks, OSS-BENCH’s “three metrics,” while simple, offer shallow coverage and fail to capture LLM weaknesses in real-world attack chains.

- The use of pass@k=1 (k=1) simplifies large-scale evaluation but overlooks generation diversity—top-tier models like Claude-3.7-Sonnet exhibit variability of 5–10% across multiple samples. Open-source models are uniformly quantized to FP16 (via Ollama), while closed-source models face no such constraints; as Table 1 shows, the leading positions are all closed-source, potentially exaggerating performance gaps (e.g., GPT-O3-Mini vs. large-scale open models). Seeds are fixed at 0 (except Claude), improving reproducibility, yet no ablation explores prompt variations: Figure 2 emphasizes “memory safety + efficiency,” but results appear sensitive to engineering choices (e.g., bug repair tasks). The model set is skewed toward mainstream 2025 releases (17 models, such as Qwen3), ignoring niche or emerging models and non-English/multilingual prompts. Regarding fairness, the chained scoring scheme (c1=s1, c2=s2×c1/100, etc.) accounts for dependency but applies default weights (w1=w2=w3=1/3 + wd=0.1) without sensitivity analysis; by contrast, CASTLE employs weighted ensembles for greater robustness, whereas OSS-BENCH risks dominance by a single metric.

- The framework is strictly limited to compiled languages (e.g., C/C++), leveraging compilation and sanitizer-based ground truth, but excludes scripting languages such as Python and JavaScript, which account for over 50% of GitHub projects according to the 2025 Stack Overflow survey. Only PHP and SQLite3 benchmarks are instantiated, and while the paper claims extensibility to projects like Nginx, Docker, or TensorFlow C-API, this requires manual adapter development (e.g., build scripts, log parsers), introducing human intervention and contradicting the core claim of “no manual or LLM involvement.” Function extraction via libclang is restricted to 10–256 token segments, filtering out ~10% of code and ignoring long functions, macros, or templates. Furthermore, multi-language hybrid projects (e.g., C++ with Python bindings) are not supported, biasing the benchmark toward infrastructure software and failing to represent mainstream domains such as web, app, and ML development. Comparable work like BUILD-BENCH has already extended automated compilation evaluation to multiple OSS projects, highlighting OSS-BENCH’s limited flexibility.

- OSS-BENCH operates at an extremely large evaluation scale—for example, OSS-BENCHphp requires each model to handle roughly 38 million tasks (N=10,534 functions recompiled + 2,000 × M ≈ 19,000 test/security checks), while OSS-BENCHsql involves about 2.41 million tasks. This demands high-end hardware (e.g., AMD EPYC 9184X CPU, NVIDIA H100 GPU, 512GB RAM) and specific environments (Ubuntu 22.04/24.04). Although the paper frames this as “intentional heavy-load standardization,” it creates a significant barrier for academia and small teams: open-source models must run via Ollama with fp16 quantization, further amplifying GPU requirements, whereas closed-source models (e.g., GPT-O1) face fewer constraints. Overall, the evaluation cycle spans weeks to months, reducing reproducibility. In contrast, benchmarks like SWE-bench rely on lightweight GitHub API calls, consuming 10–100× fewer resources. OSS-BENCH’s “million-scale tasks,” while impressive, risk being perceived as over-engineered, hindering community validation and extensibility.

**Questions:**

Please refer to the weaknesses.

---

### Official Review · Reviewer_fhyK · 2025-10-30

**Soundness:** 3
**Presentation:** 3
**Contribution:** 1
**Rating:** 2
**Confidence:** 4

**Summary:**

The paper introduces OSS-Bench, an automated benchmark for evaluating the code generation performance of LLMs on real open-source software projects. OSS-Bench extracts functions fromOSS repositories, asks LLMs to rewrite them and then tests whether the code compiles, whether it passes functional tests, and whether it avoids memory-safety issues detected by fuzzing.

**Strengths:**

Measuring the quality of LLM-generate code is important. The benchmark seems to achieve it's stated goals.

**Weaknesses:**

The novelty is very weak and issues around memorization are mentioned but not well mitigated. Currently only a few projects are supported and all are in C or C++ limiting the generality of OSS-BENCH (though there's no reason why it couldn't be extended to other compiled languages). Currently, the given correctness properties, especially memory safety, seem only applicable to C/C++ where there is weak memory safety. The evaluation is not comprehensive and only covers particular hyper-parameters, there is no sweep over different operating points for the fuzzing part for example.

**Questions:**

Why is the prompting strategy blind to the code sample being tested. I.e. prompting an LLM to improve memory safety in code that does not handle memory can result in unnecessary memory operations and potential segmentation faults. It was surprising to me that there were that many segv's from the top LLMs -- more explanation and investigation into this issue would be appreciated.

Please give more details on your fuzzer setup, how did you get/select seed inputs? The Fuzzer is PHP specific as well, how did you (or did you) test the other paplications in OSS BENCH? Why not use a generic fuzzer like AFL?

---

### Official Review · Reviewer_so3N · 2025-10-31

**Soundness:** 3
**Presentation:** 3
**Contribution:** 3
**Rating:** 6
**Confidence:** 3

**Summary:**

This paper introduces OSS-Bench, a benchmark that evaluates coding capabilities of LLMs by editing existing functions in large-scale, real-world open-source projects (PHP, SQLite), and then running a full pipeline including compilation, software testing, and memory-safety checks via sanitizers. The authors define 3 metrics and compute the scores in a chained scheme. Results on 17 models show substantial gaps between model size and real-world robustness.

**Strengths:**

1. Building and testing against mature OSS projects reduces human effort, avoids static datasets, and sidesteps LLM-as-judge issues.
2. This work emphasizes security by integrating systematic sanitizer checks and fuzzing extension.
3. This framework is extensible to other OSS projects.
4. The evaluation gives some insightful takeaways, such as model size does not guarantee memory safety in real codebases.

**Weaknesses:**

1. This paper claims to benchmark the "coding capabilities" of LLMs. However, the proposed framework primarily measures function-level editing/optimization of existing code, not code generation from natural language/specifications. This somewhat weakens the claim of "Realistic Open-Source Software Development Pipeline" since real-world software development workflows involve more code generation than editing/optimization.
2. The metrics do not capture readability, complexity, style consistency, or long-term maintainability. The Delta metric only encourages more change, not better code quality.
3. Supplying only the target function with little module/call-site context can incentivize generic defensive guards (e.g., extra NULL pointer checks and range checks) that are redundant in the wider codebase. Current metrics neither detect nor penalize such redundancy.
4. Although OSS-Bench tries to optimize the code, the evaluation lacks runtime measurements, so slower but safer edits can get more scores.

**Questions:**

1. Do you plan to add runtime performance measurements so that "optimization" is evaluated beyond compilation/tests/sanitizers? (See weakness 4)
2. Have you tried providing context when prompting the model to reduce redundant local checks? (See weakness 3)
3. How would you design and validate an "ergonomics" score to ensure edited code remains readable and maintainable over time? (See weakness 2)

---

### Meta-Review · Area_Chair_8Qbb · 2026-01-07

**Summary:**

The paper presents OSS-Bench, a benchmark for evaluating LLMs’ coding capabilities by editing functions in large-scale real-world open-source projects (e.g., PHP, SQLite) and executing a complete validation workflow (compilation, software testing, memory safety detection via code cleaning tools). The authors define 3 evaluation metrics and use a chain calculation method for scoring. Tests on 17 models reveal a significant gap between model parameter size and real-world robustness. However, reviewers identify critical flaws that invalidate the paper’s contribution, including insufficient justification for OSS-Bench’s workflow design, lack of comparative analysis with existing coding benchmarks to demonstrate uniqueness, and inadequate validation of the chain calculation method’s reliability. No author rebuttal was submitted to address these concerns. Therefore, I recommend rejecting this paper.

**Reviewer Concerns:**

The authors did not conduct a rebuttal, resulting in all the issues raised by the reviewers remaining unresolved.

**Reviewer Scores:**

Since no rebuttal was performed, the default score remains unchanged at 6, 2, 4, 4.

---

### Decision · Program_Chairs · 2026-01-26

Reject